# Therapeutic Efficacy and Mid-Term Durability of Urethral Sphincter Platelet-Rich Plasma Injections to Treat Postprostatectomy Stress Urinary Incontinence

**DOI:** 10.3390/biomedicines10092235

**Published:** 2022-09-08

**Authors:** Ping-Jui Lee, Yuan-Hong Jiang, Hann-Chorng Kuo

**Affiliations:** 1Division of Urology, Department of Surgery, Shin Kong Wu Ho-Su Memorial Hospital, Taipei 111045, Taiwan; 2Department of Urology, Hualien Tzu Chi Hospital, Buddhist Tzu Chi Medical Foundation and Tzu Chi University, Hualien 97004, Taiwan

**Keywords:** platelet-rich plasma, regenerative medicine, regeneration, urethral sphincter, postprostatectomy incontinence, stress urinary incontinence, therapeutic efficacy, durability

## Abstract

Platelet-rich plasma (PRP) is used for tissue repair and regeneration. Herein, we investigated the therapeutic efficacy and mid-term durability of injections of PRP into the urethral sphincter for the management of postprostatectomy incontinence (PPI). Thirty-nine patients with PPI that were refractory to conservative treatments were prospectively enrolled. They received repeated PRP urethral sphincter injections monthly for a total of four months. The primary endpoint was the Global Response Assessment (GRA) score after treatment. The secondary endpoints included changes in the stress urinary incontinence (SUI) visual analog scale (VAS) from baseline to the end of follow-up and urodynamic parameters from baseline to 3 months. The mean follow-up period after the entire treatment course was 21.0 ± 11.3 (range: 1.6–36.3) months. After PRP injections, the median GRA score with quartiles was 2.0 (1.0, 2.0). The SUI VAS and abdominal leak point pressure significantly improved from 6.9 ± 1.8 to 4.4 ± 2.3, *p* < 0.001, and from 74.8 ± 37.0 to 115.5 ± 57.9 cmH_2_O, *p* = 0.004, respectively, after the fourth PRP urethral sphincter injection. Following PRP urethral sphincter injections, the severity of SUI significantly reduced, indicating efficacy and mid-term durability as a novel treatment for PPI.

## 1. Introduction

Stress urinary incontinence (SUI) is characterized by an involuntary loss of urine on effort or physical exertion. When SUI occurs in men after radical prostatectomy (RP), it is referred to as postprostatectomy incontinence (PPI). PPI can cause significant distress and affect quality of life. As the number of surgeries for prostate cancer is increasing, PPI prevalence has also concomitantly increased [1].

There are several methods for the initial treatment of PPI, which include lifestyle interventions, bladder retraining, pharmacotherapy, pelvic floor muscle training, and combination therapy [2]. Surgical interventions are currently used in clinical practice to restore urethral competence, such as male slings and artificial urinary sphincters (AUSs) in unresponsive cases. Bulking agents have been proposed but have shown low efficacy, so their use is limited to patients not suitable for open surgery and in very selected cases [3,4]. Nevertheless, in real-life clinical practice, patients’ motivation and preference should also be considered. Most patients whose symptoms are not severe enough may hesitate and even refuse to undergo invasive surgery. For this reason, a simpler, less invasive but effective way is needed for patients with PPI for whom invasive surgical interventions are contraindicated and who exhibit reluctance.

Platelet-rich plasma (PRP), an autologous blood-derived product, is directly obtained from patients’ peripheral blood and is rich in platelets and a pool of cytokines, chemokines, and growth factors [5]. The wide range of secreted proteins and growth factors within the α-granules in platelets has been shown to promote thrombosis and hemostasis as well as chemotaxis, cell proliferation, differentiation, neo-angiogenesis, vascular modeling, and immune interactions [6,7]. These proteins repair damaged tissue and rejuvenate aged cells, and contribute to recovery and regeneration [5]. Therefore, PRP is an innovative and versatile formulation and has been widely used in regenerative medicine [8].

In 2016, the potential role of PRP in treating SUI was first proposed [9] and was applied in an animal model in 2019, in which a significant increase in leak point pressure was observed [10]. Based on the hypothesis that deficient urethral sphincter function improves following regenerated innervation and increased striated muscle cell volume via repeated PRP urethral sphincter injections, several recent studies have shown the safety and therapeutic efficacy of autologous PRP urethral sphincter injections for the treatment of SUI in humans [11,12,13,14].

Moreover, in terms of surgical burden and adverse events, autologous PRP urethral sphincter injections for PPI are less invasive and more advantageous compared with synthetic material implantation, such as male slings and AUSs. However, durability is yet to be investigated. Herein, we assessed the therapeutic efficacy and durability of autologous PRP urethral sphincter injections for the management of PPI that was refractory to initial conservative treatment.

## 2. Materials and Methods

### 2.1. Study Design, Setting, and Sample Size

This was a single-center, uncontrolled prospective study, conducted between September 2018 and March 2022. We aimed to assess the therapeutic efficacy and durability of autologous PRP urethral sphincter injections for the management of PPI. We determined the total sample size required was 34 for a priori analysis using G*Power software (latest ver. 3.1.9.7; Heinrich-Heine-Universität Düsseldorf, Düsseldorf, Germany).

### 2.2. Participant

Patients who exhibited de novo SUI following RP and were refractory to initial conservative treatment, such as lifestyle modifications, bladder retraining, pelvic floor muscle training, and pharmacotherapy, for at least 12 months were enrolled. The primary etiology of PPI, i.e., urethral incompetence, needed to be fulfilled. Patients with a large post-void residual (PVR) volume ≥ 150 mL, known platelet dysfunction, anticoagulant use, critical thrombocytopenia, hypofibrinogenemia, hemodynamic instability, sepsis, acute or chronic infections, chronic liver disease, and known malignancy outside the urinary tract were excluded from the study [12].

The following clinical data were collected: age, duration of SUI, systemic comorbidities, RP method, and total follow-up period after treatment (in months). The SUI visual analog scale (VAS) [11] (Table 1), which was designed according to the Stamey SUI grading system [15], was used to determine the severity of PPI in this study. The SUI VAS score ranged from 0 to 10 and was reported by patients’ subjective perception of incontinence condition, where 0 indicates complete continence, and 10 indicates total urinary incontinence.

All eligible patients underwent videourodynamic studies (VUDSs) and abdominal leak point pressure (ALPP) measurement at the beginning of the study. The procedures of these studies and the terminology of the urodynamic parameters were performed in accordance with the recommendations of the International Continence Society (Abrams et al., 2003). The urodynamic parameters, including cystometric bladder capacity (CBC), voided volume (Vol), PVR, maximum flow rate (Qmax), detrusor pressure at Qmax (Pdet.Qmax), and ALPP were obtained during the exam. The bladder outlet obstruction index (BOOI, defined as Pdet.Qmax–2 Qmax) and corrected maximum flow rate (cQmax; defined as Qmax/ Vol^1/2^) were calculated as well.

### 2.3. Preparation of PRP and Urethral Sphincter Injection Procedure

PRP was prepared using 50 mL of whole blood drawn from the peripheral vein and delivered to the central laboratory of the hospital for two centrifugation steps. First, to separate the plasma and erythrocyte layers, a soft spin at 200× *g* for 20 min at 20 °C was performed. The upper plasma layer was aseptically collected and subjected to a second hard spin at 2000× *g* for 20 min at 20 °C. We obtained 5 mL of PRP, and the platelet concentration of both the PRP product and whole blood was recorded.

Given the mini-invasiveness nature of the procedure, it was performed under light sedation by the anesthesiologist. The PRP was circumferentially injected into the external urethral sphincter at five injection sites using a rigid cystoscopic injection instrument through the urethra (23 Fr; Richard Wolf, Knittlingen, Germany). To confirm the effects following each injection, swelling of the urethral surface was identified [12]. Patients did not undergo urethral catheterization and were discharged shortly after the injections, provided no complications developed. The procedures were repeated on a monthly basis for a total of four injections.

### 2.4. Endpoints

All patients were requested to complete questionnaires and an interview at baseline, one month after each urethral PRP injection (i.e., just before the next injection), and at monthly return visits after the treatment course. Treatment outcomes, injection-related complications, and postoperative voiding status were thoroughly recorded. To evaluate changes in the lower urinary tract function, VUDSs and ALPP measurement were repeated three months following the fourth injection.

Treatment outcomes were assessed using the self-reported SUI VAS [11] after each injection and the Global Response Assessment (GRA) score (categorized from −3 to 3, which indicate markedly worsened to markedly improved condition, respectively) [16] after the final injection. The post-treatment GRA score was the primary endpoint of this study, and a successful outcome was defined as a GRA score ≥ 2 (moderate and marked improvement), whereas clinical improvement was defined as a GRA score ≥ 1. The secondary endpoints included changes in the SUI VAS from baseline until the end of follow-up and urodynamic parameters from baseline to three months following the final injection. Additionally, we assessed predictive factors for successful treatment outcomes.

### 2.5. Statistical Analysis

We used SPSS version 25.0 (IBM Corp. Released 2017. IBM SPSS Statistics for Windows, Version 25.0. Armonk, NY: IBM Corp.) for statistical analyses. The categorical data are represented as numbers and percentages, whereas the continuous variables are represented as means with standard deviations and as medians with quartiles. To evaluate the therapeutic effects, we used Wilcoxon’s signed-rank test to distinguish the difference in variables in patients between baseline and after treatment and Wilcoxon’s rank sum test for statistical comparisons of variables in the between-subgroup analysis. Logistic regression analysis was used for the prediction of factors associated with successful outcomes. *p* < 0.05 was considered statistically significant.

If no urine leak in response to coughing or Valsalva maneuver was observed in VUDSs and ALPP measurements, no ALPP value was recorded. Furthermore, once a patient received additional anti-incontinence management during follow-up, the SUI VAS after that time point was not adopted for analysis.

## 3. Results

At data cutoff on 20 March 2022, we reported an additional 18 months of follow-up from the initial analysis [12] and 11 more eligible patients were recruited. Thirty-nine patients with a mean age of 71.6 ± 8.1 (range: 53–87) years and a mean SUI duration of 52.8 ± 44.1 (range: 12.2–189.5) months were enrolled in this study. Open, laparoscopic, and robotic-assisted RP were performed in 7, 7, and 25 patients, respectively. A total of 11 (28.2%) patients presented with an SUI VAS score of 1–5 (i.e., Stamey Grade 1 SUI), 25 (64.1%) presented with an SUI VAS score of 6–9 (i.e., Stamey Grade 2 SUI), and 3 (7.7%) presented with an SUI VAS score of 10 (i.e., Stamey Grade 3 SUI). Pad protection was necessary in all patients before treatment. The mean concentration ratio of PRP to the whole blood of each injection was 3.4 ± 1.2, 3.4 ± 1.0, 3.6 ± 1.2, and 3.5 ± 1.2. The mean follow-up period after the entire treatment course was 21.0 ± 11.3 (range: 1.6–36.3) months.

After four PRP injections into the urethral sphincter, the median GRA score with quartiles was 2.0 (1.0, 2.0) (mean with standard deviations: 1.7 ± 0.9, range: −1–3). Overall, 23 (58.9%) patients achieved a successful outcome (GRA score ≥ 2), 36 (92.3%) experienced clinical improvement (GRA score ≥ 1), 2 (7.7%) revealed no improvement (GRA score = 0), and 1 was slightly worse than before treatment (GRA score = −1) (Table 2). The SUI severity was significantly reduced one month after the fourth PRP urethral sphincter injection (SUI VAS from 6.9 ± 1.8 to 4.4 ± 2.3, *p* < 0.001) (Figure 1A). The greater reduction in the SUI VAS score was associated with a better treatment outcome, as defined by the GRA score (Pearson correlation coefficient = 0.568), and this correlation was significant (*p* < 0.001). During the treatment course, therapeutic effects were observed immediately after the first injection and were consolidated by repeated injections. However, the treatment effect was noted to gradually decline 24 months after the fourth PRP injection during follow-up.

According to the subgroup analysis, the changes in the SUI VAS score after four urethral sphincter PRP injections were significant in cases that were both successful and unsuccessful (all *p* < 0.001) in the beginning. However, changes were not significant in the unsuccessful subgroup after follow-up (Figure 1B). Sixteen patients demonstrated no urine leakage in post-treatment stress tests during VUDS, and none of the urodynamic parameters significantly changed except for ALPP (increased from 74.8 ± 37.0 to 115.5 ± 57.9 cmH_2_O, *p* = 0.004) (Table 2). PRP injections through the urethral sphincter for PPI did not result in any negative effects on lower urinary tract function. As illustrated in Table 3, the logistic regression model revealed that all potential predictive factors, including age, duration of SUI, initial SUI severity, pretreatment lower urinary tract function, and platelet concentration, did not have a significant impact on patients’ outcomes.

There were no major complications reported, and in no case did the incontinence profile of the patient deteriorate compared with before treatment. Mild hematuria and painful micturition perioperatively occurred in three (7.7%) patients, and abdominal straining to void was reported in one (2.6%) patient during the follow-up period. All of the above symptoms resolved after the administration of conservative treatment. Otherwise, no clinically significant adverse events or complications, such as acute urinary retention or symptomatic urinary tract infection, developed.

Finally, 3 (7.7%) patients achieved complete continence and no longer needed safety pads, 17 (43.6%) were satisfied with the existing state despite occasional incontinence, 8 (20.5%) were satisfied with current incontinence improvement but would receive additional PRP injections to enhance the therapeutic efficacy, and 6 (15.4%) were not satisfied with current incontinence condition and planned to receive further intervention. The other five (12.8%) patients underwent additional anti-incontinence surgery, including three (7.7%) with a male sling and two (5.1%) with an AUS.

## 4. Discussion

Based on significant clinical and urodynamic evidence, this study demonstrated the efficacy, safety, and durability over the medium term of repeated PRP injections into the urethral sphincter for the treatment of PPI. Thirty-six (89.7%) patients reported a positive response to treatment, and twenty-three (58.9%) achieved successful outcomes. The SUI severity was significantly reduced until the end of the follow-up period, regardless of baseline characteristics or platelet concentration after repeated urethral sphincter PRP injections. Moreover, ALPP, which indicates urethral resistance, significantly improved after treatment. Accordingly, in terms of concerns regarding efficacy, safety, and durability, urethral sphincter PRP injection may be considered an initial treatment for patients with PPI, especially when the SUI severity is mild to moderate and patients have resisted surgical intervention after previous radical prostatectomy.

According to the recommendations of the International Consultation on Incontinence, medically refractory PPI generally involves invasive surgical interventions, and AUS is possibly the most effective and gold standard treatment option [4,17]. However, it is not suitable for all patients given the associated costs, complications, and substantial revision rate. Male slings are an alternative for AUS, but there is insufficient long-term data, and the success rate shows a wide variation (9–87%) given the different sling types. Injectable bulking agents, the effects of which deteriorate over time and offer very low cure rates, are an inferior option that should only be utilized when more effective options are contraindicated [17].

Possible clinical utilization of stem cells in onco-urology and regenerative urology has been previously investigated and well-reviewed [18]. Using cell-based therapeutic approaches, regenerative medicine may allow the improvement in the external (striated muscle) and internal (smooth muscle) urethral sphincter muscle function, neuromuscular transmission, and blood supply [19,20]. Recently, different types of stem cells, such as skeletal-muscle-derived stem cells, bone marrow stem cells, human umbilical mononuclear cells, adipose-derived stem cells, and modified/sorted stem cells, have been used for SUI treatment and have been shown to be feasible and safe in the short term [21,22]. According to the regenerative capacity of PRP, in our previous studies, we successfully resumed and increased urethral competence through repeated PRP urethral sphincter injections, which improved SUI severity in patients with non-neurogenic intrinsic sphincter deficiency as well as in female patients with SUI [11,14].

In our most recent study, we showed that the therapeutic effects of repeated PRP urethral injections are safe and effective and that PPI injection was a novel approach for the management of PPI in 28 patients. In a urodynamic study, most patients showed significant improvement in the severity of SUI as well as in ALPP. Additionally, this repeated PRP urethral sphincter injection therapy for PPI did not result in adverse effects on lower urinary tract function [12]. Despite the prominent results, a long-term follow-up was lacking, and thus, some doubted the durability of the treatment outcomes. In the present study, we investigated the mid-term durability of PRP urethral sphincter injection for the treatment of PPI. As shown by the results, the clinical therapeutic effect did not dissipate before 18 months after surgery, which dispels the false belief of the temporary bulking effect of urethral PRP injection.

Women with SUI who received PRP injections achieved a slightly higher treatment success rate, as defined by cure and improved symptoms, in the younger group [13]. The results herein and from our previous studies are not concurrent with previous findings in the reported literature, as the influence of age on clinical outcomes was not significant [12,14]. Moreover, we could not identify significant correlations among SUI duration, initial SUI severity, lower urinary tract function, platelet concentration, or clinical outcomes. Similarly, thus far, a correlation between a “more is better” approach for the use of higher platelet concentrations and improved healing and patient outcomes has not been supported [23]. This study, with a small number of cases, revealed that most patients with PPI, regardless of their baseline characteristics, can benefit from urethral injection of PRP.

The lack of randomization and placebo control groups constitute the limitations of our study. While PRP is becoming a greater research focus, few randomized controlled studies have been performed, especially those that consider ethical conflicts. A head-to-head comparative study to compare urethral PRP injections with different methodologies, such as platelet-poor plasma or urethral bulking agent, may be needed to determine definite therapeutic efficacy in future studies. The other important consideration is platelet concentration. While platelet concentrations that are two to three times larger than the mean in other studies have been widely used [24], more research on the potential different effects of PRP with varying platelet concentrations is needed. Third, although the SUI severity score in the present study was established based on the well-accepted Stamey SUI grade, it is yet to be fully validated. Finally, this study provided evidence, such as subjective symptom scores and objective ALPP improvement, based on urodynamic examinations. However, direct evidence to validate the supposed recovery of urethral sphincter muscle function, neuromuscular transmission, and blood supply via repeated PRP urethral sphincter injections is still lacking. Application of urethral sphincter imaging (ultrasound and/or MRI) or histological investigation may be one way to quantify the cellular changes at the injection site. Further studies will reveal the role of PRP as a potential treatment option for stress urinary incontinence. Because PRP urethral sphincter injection for PPI is a novel regenerative medicine treatment, the duration of therapeutic efficacy has not been tested. This study provides evidence that urethral PRP injection might not be life-long durable, and repeat urethral injection might be necessary in patients who experience faded therapeutic effect over time.

## 5. Conclusions

PRP injection into the urethral sphincter in patients with PPI refractory to conservative treatment after 12 months from radical prostatectomy is a safe, well-tolerated, minimally invasive procedure, and seems to have good efficacy in the medium term, as demonstrated by significant clinical and urodynamic evidence. It can be proposed as an initial approach in patients not motivated for invasive treatments or with significant comorbidities and unfit for surgery. More evidence on the effects of PRP injections at the cellular level in the urethra needs to be collected in the near future.

## Figures and Tables

**Figure 1 biomedicines-10-02235-f001:**
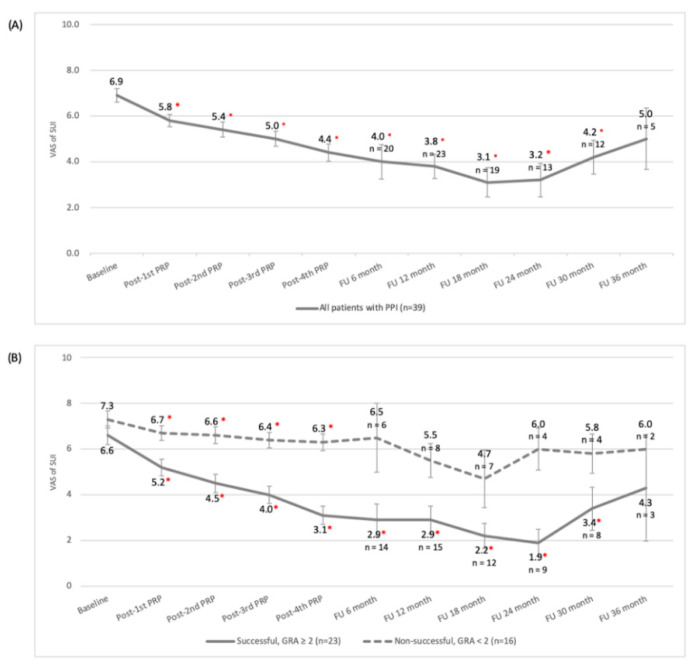
Changes in stress urinary incontinence (SUI) severity in accordance with the visual analog scale of SUI after autologous platelet-rich plasma urethral sphincter injections (**A**) in all patients; (**B**) subgroup analysis by outcome. Note: 1. Data are presented as the mean and standard error of the mean (SEM). 2. Patient dropped out after receiving further anti-incontinence treatment. 3. Abbreviation: FU = follow-up. * *p* value < 0.05 (compared with baseline).

**Table 1 biomedicines-10-02235-t001:** Visual analogue scale (VAS) for assessment of stress urinary incontinence (SUI).

NRS of SUI	Corresponding Stamey Grade of SUI	Situation of SUI	Frequency of SUI (on Situation)	Pad Protection
0	0	Complete dryness	0	No
1	1	Any mild or severe straining situation	1 episode per days to weeks	No/Yes
2	1	Heavy straining/squatting	1 episode per day	No/Yes
3	1–2	Coughing/ sneezing/laughing/nighttime/change position	1 episode per day	No/Yes
4	1	Heavy straining/squatting	>1 episode per day	No/Yes
5	1–2	Coughing/ sneezing/laughing/nighttime/change position	>1 episode per day	No/Yes
6	2	Walking	<50% situation (every day) and >1 episode per day	Yes
7	2	Very mild movement/change position	<50% situation (every day) and >1 episode per day	Yes
8	2	Walking	≥50% situation (every day) and >1 episode per day	Yes
9	2	Very mild movement/change position	≥50% situation (every day) and >1 episode per day	Yes
10	3	Any condition; persistent	All the time	Yes

**Table 2 biomedicines-10-02235-t002:** Clinical outcomes and change in urodynamic parameters after platelet-rich plasma urethral sphincter injection treatments.

	Total (*n* = 39)		
GRA score**,** median (Q1, Q3)	2.0 (1.0, 2.0)		
GRA score = −1	1 (2.6%)		
GRA score = 0	2 (5.1%)		
GRA score = 1	13 (33.3%)		
GRA score = 2	16 (41.0%)		
GRA score = 3	7 (17.9%)		
Improvement (GRA score ≥1)	26 (89.7%)		
Successful outcome (GRA score ≥ 2)	23 (58.9%)		
	Baseline	After treatment	*p*-value
VUDS parameters**,** mean ± SD			
CBC (mL)	287.6 ± 115.1	297.3 ± 160.9	0.746
Vol (mL)	257.6 ± 112.7	260.8 ± 136.1	0.881
PVR (mL)	32.1 ± 75.0	35.8 ± 142.0	0.872
Qmax (mL/s)	11.0 ± 5.0	11.7 ± 5.0	0.443
cQmax (Qmax/Vol^1/2^)	0.67 ± 0.29	0.72 ± 0.32	0.356
Pdet.Qmax (cmH_2_O)	14.8 ± 10.2	13.1 ± 9.9	0.212
BOOI	−6.94 ± 14.2	−9.6 ± 14.3	0.309
ALPP (cmH_2_O) §	74.8 ± 37.0	115.5 ± 57.9	0.004

ALPP: abdominal leak point pressure; BOOI: bladder outlet obstruction index; CBC: cystometric bladder capacity; GRA: global response assessment; Pdet.Qmax: detrusor pressure at maximum flow rate; PVR: postvoid residual; Qmax: maximum flow rate; VAS of SUI: visual analogue scale of stress urinary incontinence severity; Vol: voided volume; VUDS: videourodynamic study. § Twelve patients in the pretreatment stress test, and sixteen patients in post-treatment stress tests during VUDS did not demonstrate urinary leakage (i.e., no valuable ALPP), and the ALPP values of these patients were excluded for analysis.

**Table 3 biomedicines-10-02235-t003:** Logistic regression analysis for predictors of successful outcome.

	Unadjusted Odds Ratio (95% CI)	*p*-Value
Age	1.012 (0.935–1.096)	0.764
SUI duration	1.009 (0.991–1.027)	0.345
Initial SUI severity		
VAS of SUI 1–5	Reference	-
VAS of SUI 6–10	0.417 (0.072–2.412)	0.329
Baseline VUDS parameters		
CBC	1.003 (0.998–1.009)	0.265
Vol	1.001 (0.994–1.008)	0.774
PVR	1.008 (0.994–1.022)	0.281
Qmax	0.956 (0.859–1.065)	0.415
cQmax	0.299 (0.044–2.039)	0.218
Pdet.Qmax	0.955 (0.897–1.016)	0.148
BOOI	0.988 (0.941–1.036)	0.611
ALPP	1.019 (0.995–1.043)	0.126
Platelet concentration ratio		
1st injection	0.981 (0.582–1.652)	0.942
2nd injection	1.001 (0.996–1.007)	0.668
3rd injection	1.793 (0.967–3.325)	0.064
4th injection	1.496 (0.852–2.625)	0.161

ALPP: abdominal leak point pressure; BOOI: bladder outlet obstruction index; CBC: cystometric bladder capacity; Pdet.Qmax: detrusor pressure at maximum flow rate; PVR: postvoid residual; Qmax: maximum flow rate; VAS of SUI: visual analogue scale of stress urinary incontinence severity; Vol: voided volume; VUDS: videourodynamic study.

## Data Availability

Data is contained within the article.

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
