# Peer review of "Therapeutic Efficacy and Mid-Term Durability of Urethral Sphincter Platelet-Rich Plasma Injections to Treat Postprostatectomy Stress Urinary Incontinence"

_biomedicines, 2022, doi:10.3390/biomedicines10092235_

Round 1

Reviewer 1 Report

This is a good-sized study considering the novelty of the method. It suffers from the lack of a placebo arm, however results are of interest as it is a real-life study with a longer follow-up than the previously published ones.

The manuscript is mostly well written. Some points may need to be addressed to make it clearer to the reader:

1.      In Abstract, lines 16-17, “They received four PRP urethral sphincter injections on a monthly basis” actually could be meaning that patients received 4 injections per month. Please revise.

2.     2.  In Results section, the results presented in lines 205-211 appear to be somewhat confusing or contradicting the results presented in lines 175-178. Although lines 205-211 refer mostly to treatment satisfaction as opposed to clinical improvement, lines 207-211 are particularly confusing: “6 patients (17.6%) planned further intervention”: what does ‘further intervention’ refer to?  In the same paragraph the authors report that 5 patients received additional surgery (lines 208-211). Are these patients different to those who planned further intervention? Also, “8 (20.5%) confirmed a clinical therapeutic effect and received the extra injection course”. What is the extra injection course? Were there more than the 4 protocol injections? Or only 8 out of 39 patients received all 4 injections of the study protocol? Also, what is considered a ‘clinical therapeutic effect’ in this paragraph? Finally, if only 8 patients confirmed a clinical therapeutic effect, how does this match with the 23 patients who achieved a successful outcome (line 175) or the 36 patients who reported clinical improvement (line 176)? Please clarify / revise.  

3.      3. Since this was a prospective study, it should be mentioned whether the protocol underwent ethical approval and patients signed a written informed consent. Also, was the study registered in any clinical trial database?

Reviewer 2 Report

This manuscript is interesting and offers some valuable data in the field of non-invasive treatment of urinary incontinence after radical surgery for prostate cancer. However, it needs a number of changes before being accepted for publication in Biomedicines. The changes needed are detailed in the text below.

TITLE: The most important finding of this paper, which is also its main strength, is that the treatment proposed by the Authors is effective and durable at the mid-term follow-up. This is the real finding. I think therefore that this fact should be clearly stated in the title of the paper, which is at the moment a bit generic and not very attention-catching. A proposed change for the title is:

Therapeutic efficacy and mid-term durability of urethral sphincter platelet-rich plasma injections to treat post-prostatectomy stress urinary incontinence

ABSTRACT:

Line 28: “mid-term durability”. You should always refer to “mid-term durability” in the text, because this is what you are investigating. You are showing no results or data on long-term durability of this procedure

INTRODUCTION:

General comment:

Do not be superficial on the clinical part of the problem. Despite the current amount of data from clinical research in the last years on stem cells and regenerative medicine in treating urinary incontinence, large-scale transfer of data from bench to bedside is still lacking. Possible clinical utilization of stem cells in Urology have been investigated and well reviewed in Mancini M. et al, Urologia Journal, 2016, a citation that should be added to the references (https://doi.org/10.5301/uro.5000165).

Your study presents some good new findings on PRP injection in the urethral sphincter with a potential clinical value, and for designing personalized approaches to treatment, based also on patients’ motivation and preference. Stress better these concepts in the Introduction, Discussion and Conclusions (some examples are offered in the texts below). Be precise on the clinical facts and on the clinical conceptualization (as shown below).

Line 43: Surgical interventions in unresponsive cases are currently used in clinical practice, such as urethral slings and artificial sphincters, in different clinical cases. Bulking agents have been proposed but have shown low efficacy, so at the moment their use is limited to patients not suitable for open surgery and in very selected clinical cases.

Line 48-50: Re-phrase the concept. You cannot put together bulking agents, slings and sphincters as if they were comparable alternative measures. You should clearly state that surgery, such as urethral slings of artificial sphincters are still the gold standard for treatment of PPI. Non-surgical procedures have been proposed and can be tried in patients reluctant to undergo surgery or in whom invasive surgical interventions are contra-indicated, until more definitive data on their efficacy and durability are provided.

Line 54: has been shown to promote

Line 65: several recent studies

Line 68-70: Same as line 48-50

Line 122: Explain why general anesthesia was necessary. Maybe you could add here that, given the mini-invasiveness of the procedure, local anesthesia plus mild sedation or could be also tried, to reduce medicalization of the patients.

RESULTS

General comment: here you have to clearly show/prove that your proposed strategy is safe and effective and durable in the mid-term. This is a very difficult but highly clinically significant task.

Line 185: “the treatment effect declined gradually over time. The SUI severity was reduced at 1 month after the fourth injection”. I think this concept needs to be clarified better for the readers. How long did the benefits last? Why do the Authors think they declined? Could it be hypothesized to continue the injections and try a second course?

Line 193: This needs to be stressed better. I think you have to better underline the safety profile of the procedure. Also, you should say that there was no major complication, and in no case the incontinence profile of the patient was worse than before treatment. This is an important point.

Line 195: The predictive factors had no influence on the effect of treatment. This is quite a striking result. What about hormonal treatment/radiation therapy? Any patients submitted to radiation therapy after radical prostatectomy and before treatment with PRP? Radiation therapy has been shown to have a detrimental effect on PPI treatment. Can you add anything on this regard? It would be very interesting to know if this is the case also in case of PRP injections. You could speculate, if you have any specific result, that PRP treatment could not be affected by previous radiation therapy, since it is based on the effect of proteins/stem cells contained in the injected solution, and not so much on the quality of the cells located in the sphincter/urethra.

DISCUSSION:

General comment: The discussion section is the place where the results are compared to what has been published before. Here you must stress the strength and the originality of your work.

Here I would state again the importance of stem cell research in Urology, both in Oncology and in Regenerative Urology (well reviewed in: https://doi.org/10.5301/uro.5000165).

State clearly that AUS is the gold standard treatment for PPI, especially in severe cases. Male sling procedures are well established alternatives, especially in mild-moderate incontinence. In this clear scenario, PRP treatment may be proposed as a good initial option, offering a mini-invasive alternative with a good safety profile and acceptable mid-term results in terms of durability. The clinical therapeutic effect of PRP did not dissipate indeed before 18 months after the last injection of the therapeutic cycle in your study. It seems from the Authors results that most patients with PPI, regardless of their baseline characteristics can benefit from urethral injections of PRP in the mid-term.

Regarding the limitations of your study, I would not say that lack of randomization is one of them. The real limitation of your study is lack of a control group. The real control group would be a group of patients with the same baseline characteristics in which you perform urethral injection of the same amount of fluid /serum, without platelets and other plasma components. This would be a real control, in which you could assess, in a future study, the placebo effect of the procedure, and the therapeutic effect of injecting a fluid solution in the sphincter. Another possible strategy would be to randomize your treatment versus injection of a bulking agent in the urethra, and see what would be the difference and short and mid-term follow-up. This would be very interesting and you could possibly speculate on future studies on this line.

Last limitation: there is no assessment of cellular effects in the sphincter/urethra of your treatment. You could speculate on possible utilization of imaging (ultrasound/MRI) of the urethra or other modalities able to quantify the cellular changes on the injection site. This would be the only way to really prove your hypothesis, which is that the deficient urethral sphincter function improves following regenerated innervation and increased striated muscle cell volume via repeated PRP urethral sphincter injections.

CONCLUSIONS

Do not jump to the Conclusions that urethral sphincter PRP injections may be considered an initial treatment for patients with PPI. What you really have demonstrated is that ALPP significantly improves after treatment at the observed time point, but the effects decline over time.

Following what has been conceptualized above, I would re-phrase the Conclusions as follows:

PRP injections into the urethral sphincter in patients with PPI refractory to conservative treatment after 12 months from radical prostatectomy is a safe, well tolerated, minimally invasive procedure, and seem to have good efficacy in the mid-term. While more studies are needed to better characterize this technique, it could be proposed as an initial approach in patients not motivated for invasive treatments or with significant co-morbidities and unfit for surgery. More evidence on the effects of PRP injections at the cellular level in the urethra needs to be collected in the next future.

Round 2

Reviewer 2 Report

The paper has been improved significantly. Some minor English language spell check can be corrected during final text editing.

The Authors have done a good job in revising the text and the conceptualization